# Ideas and perspectives: Microorganisms in the air through the lenses of atmospheric chemistry and microphysics

Barbara Ervens[1], Pierre Amato[1], Kifle Aregahegn[1,2], Muriel Joly[1], Amina Khaled[1],
Tiphaine Labed-Veydert[1], Frédéric Mathonat[1], Leslie Nuñez López[1], Raphaëlle Péguilhan[1,3], and
Minghui Zhang[1,4]

[1]University Clermont Auvergne, CNRS, Institute of Chemistry Clermont-Ferrand, 63000 Clermont-Ferrand, France.
[2]now at: Department of Chemistry, York University, Toronto, ON, Canada.
[3]now at: Department of Chemical and Biochemical Engineering, Technical University of Denmark, 2800 Kgs. Lyngby, Denmark.
[4]now at: Minerva Research Group, Max Planck Institute for Chemistry, 55128 Mainz, Germany.

**Correspondence:** Barbara Ervens (barbara.ervens@uca.fr)

**Abstract.** Microorganisms in the atmosphere comprise a small fraction of the Earth' microbiome. A significant portion of this aeromicrobiome consists of bacteria that typically remain airborne for a few days before being deposited. Unlike bacteria in other spheres (e.g., litho-, hydro-, phyllo-, cryospheres), atmospheric bacteria are aerosolized, residing in individual particles and separated from each other. In the atmosphere, bacteria encounter chemical and physical conditions that affect their stress levels and survival. This article goes beyond previous overviews by placing these conditions in the context of fundamental chemical and microphysical concepts related to atmospheric aerosols. We provide ranges of water amounts surrounding bacterial cells both inside and outside clouds and suggest that the small volumes of individual cloud droplets lead to nutrient and oxidant limitations. This may result in greater nutrient limitation but lower oxidative stress in clouds than previously thought. Various chemical and microphysical factors may enhance or reduce microbial stress (e.g., oxidative, osmotic, UV-induced), affecting the functioning and survival of atmospheric bacteria. We illustrate that these factors could impact stress levels under polluted conditions, indicating that conclusions about the role of pollutants in directly causing changes to microbial abundance can be erroneous. The perspectives presented here aim to motivate future experimental and modeling studies to disentangle the complex interplay of chemical and microphysical factors with the atmospheric microbiome. Such studies will help to comprehensively characterize the role of the atmosphere in modifying the Earth' microbiome, which regulates the stability of global ecosystems and biodiversity.

## 1 Introduction

Microorganisms are the most abundant organisms on Earth and comprise about 15% of the total biological mass (Bar-On et al., 2018). This 'unseen majority' of life (Whitman et al., 1998) is an essential actor of ecosystem functioning, biogeochemical nutrient cycling, soil formation, and decomposition processes, and crucial for preserving the health of the planet (Cardinale et al., 2012). Diversity within microbial communities usually leads to higher resilience towards environmental disturbances, such as pollution, climate change, and may beneficial to more robust and healthy systems (Robinson and Breed, 2023).

A significant portion of Earth's microbiome consists of bacteria; Figure 1 provides an overview of the bacteria concentrations and abundance in major aquatic and terrestrial 'spheres' of the Earth. The highest (average) bacteria cell concentrations are found in soil ($10^{15}$ cells m$^{-3}$). The concentrations that are lower by several orders of magnitude in vegetation (phyllosphere), water (hydrosphere) and ice (cryosphere). The surfaces of these spheres are connected by the atmosphere that harbors significantly smaller cell concentrations, both related to air volume ($\sim$$10^4$ cells m$^{-3}$) and in absolute numbers ($\sim 10^{19}$ cells), which comprise an apparently negligible fraction ($< 10^{-9}$ %) of the total Earth microbiome ($> 10^{30}$ cells).

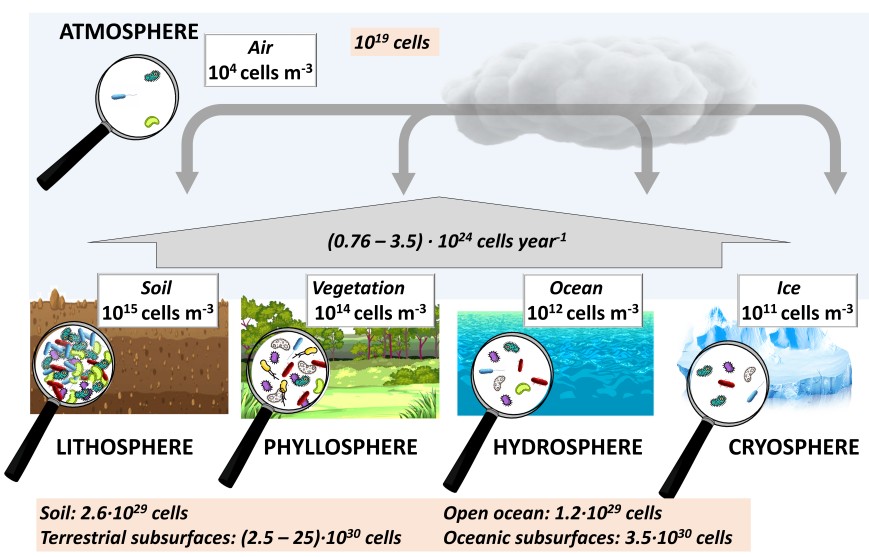

**Figure 1.** Schematic of bacteria cell concentrations in the spheres of the Earth. References: soil (prokaryotes) (Whitman et al., 1998), vegetation (Lindow and Brandl, 2003), assuming leaf volume-area ratio of $\sim$500 cm$^{-3}$m$^{-2}$ (Poorter et al., 2009), surface waters (prokaryotes): (Whitman et al., 1998), sea ice (Boetius et al., 2015), atmosphere (Burrows et al., 2009b); the values in the orange box at the bottom are from Whitman et al. (1998), emissions to the atmosphere from Burrows et al. (2009a), atmospheric cell number from Šantl-Temkiv et al. (2022).

In comparison to the microbially more densely populated spheres, the atmospheric microbiome ('aeromicrobiome') is relatively poorly characterized. In the past, the main focus on studies of airborne microorganisms addressed their role as pathogens (Pasteur, 1861). The rapid air movements and emission/deposition cycles in the highly dynamic atmosphere lead to efficient transport and displacement of microorganisms. While close to the ground, the atmospheric microbial diversity resembles that of the underlying surface ('footprint'), air masses mix at higher altitudes resulting in complex mixtures of microbial populations. Microbes follow major air movements along 'microbial highways'. However, the exact patterns of such aerobiological trajectories cannot be resolved to date since samples are relatively sparse (Smith et al., 2018). A fundamental difference in the atmosphere - as opposed to microbial environments in other Earth' parts - is the fact that microorganisms are aerosolized, i.e., they are surrounded by a finite hydration shell limiting their access to nutrients and water. Aerosol particles are continuously exposed to light, trace gases, oxidants and other chemical pollutants, which leads to unique conditions that rapidly change

and may expose the microorganisms to different levels of stress on a variety of spatial and temporal scales. These particular conditions, that greatly differ from those in the (more) homogeneous aquatic and terrestrial environments, are usually not taken into account in atmospheric microbiological studies.

The present article fills this gap by providing a new perspective on temporal and spatial scales of atmospheric chemical and microphysical factors and processes that may affect the atmospheric microbiome. In Section 2, we discuss various microscale (physico)chemical aerosol properties relevant for the atmospheric bacterial environments. (All underlying fundamental equations and parameters are summarized in the supplemental information.) Each subsection is introduced by a question we neither attempt nor intend to comprehensively answer; instead, these questions are posed to motivate future studies to explore factors that potentially control microbial stress, survival and diversity in the atmosphere. Section 3 places the preceding considerations into the context of atmospheric scenarios, specifying various environmental factors and their potential role for microbial stress under polluted and/or cloudy conditions. In the concluding Section 4, we give recommendations for future studies to advance our understanding of the atmosphere in shaping the microbiome. We point out the need of interdisciplinary efforts merging atmospheric (aerosol) sciences, microbial ecology and aerobiology.

## 2 Atmospheric bacteria: temporal and spatial scales

### 2.1 Aerosolized bacteria: does social distancing between cells matter for their functioning?

The formation of agglomerates and/or biofilms in aquatic and terrestrial habitats provides multiple advantages to bacteria in terms of protection and collective resources. Such social traits enable microbes to adapt to diverse environments, optimize resource utilization, and enhance their survival and reproductive success. Bacteria in the atmosphere are detached from such community structures as they can only be airborne upon aerosolization. Bacteria-containing particles contain a single - or at most a few - bacteria cell(s). The presence of more than a single cell in a particle leads to a larger particle. However, the resulting total particle surface area might not scale proportionally with the number of cells since the particle shape and total volume might be mostly determined by the hydration shell. In addition to gravitational settling, other particle properties and processes affect the atmospheric residence time, including horizontal transport and the ability to act as cloud condensation or ice nuclei. Burrows et al. (2009a) demonstrated that bacteria acting as cloud condensation nuclei (CCN) are generally deposited faster and, thus, have a global atmospheric residence time that is approximately half as long as bacteria not activated into droplets. Therefore, the distances that CCN- and/or IN-active bacteria travel in the atmosphere are generally comparably short.

Atmospheric concentrations of bacteria cells are typically in the range of 0.001 - 0.1 cells $cm_{air}^{-3}$ (Burrows et al., 2009a; Després et al., 2012) with typical sizes on the order of 100 nm - 1 $\mu$m (Sattler et al., 2001; Pöschl and Shiraiwa, 2015). The total atmospheric number concentration of aerosol particles of such sizes ('fine particles') ranges from $10^3$ - $10^5$ particles $cm_{air}^{-3}$ (Seinfeld and Pandis, 2006). The comparison of these numbers reveals that bacteria comprise $\ll$1% of all atmospheric aerosol particles. A cloud droplet forms by water vapor condensation on an individual particle, i.e., on a cloud condensation nucleus (CCN) that is typically in the size range of fine particles. Since single bacterial cells are often similar in size to these particles,

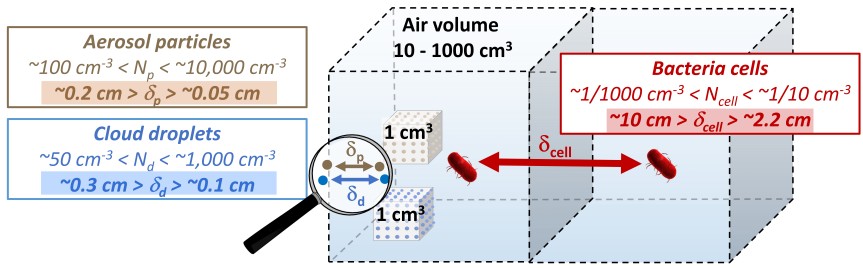

**Figure 2.** Schematic illustration of the distance between bacteria cells vs aerosol particles and cloud droplets ($\delta_{cell}$, $\delta_p$, $\delta_{dr}$) for typical atmospheric concentrations (N [cm$^{-3}$]); the distance $\delta$ [cm] corresponds to $N^{-1/3}$ (for calculation and wider ranges, cf Section S1.2 in the supplemental information).

it is likely that each particle hosts only one cell (Fankhauser et al., 2019). The fact that the bacteria number concentration is much smaller than the total CCN concentration in the atmosphere led Fankhauser et al. (2019) and Ervens and Amato (2020) to conclude that only 1 out of $\sim$10000 cloud droplets contains a bacteria cell. Zhang et al. (2021) showed that such low CCN number concentrations likely have a negligible effect on the formation and properties of warm clouds. However, in the same

study, it was pointed out that information on the hygroscopic properties of bacteria is essential as it determines the volume of the aquatic environment that is important for microbial growth, survival, and functioning (Section 2.2).

The low bacteria concentrations in the atmosphere imply that bacteria cells are separated by considerable distances. Figure 2 shows schematically the average distance between cells ($\delta_{cell}$) for typical atmospheric bacteria cell concentrations (0.001 cm$^{-3}$ < $N_{cell}$ < 0.1 cm$^{-3}$) (Section S1.2 in the supplemental information). The schematic shows that bacteria can be expected to be

separated on average by several centimeters ($\sim$2.2 - 10 cm), whereas other aerosol particles or cloud droplets are apart by several millimeters ($\delta_p$, $\delta_{dr}$). This social distancing of bacteria impairs the collective traits and functions that bacteria can benefit in other environments (Ross and Whiteley, 2020). In addition to such mutualistic behavior, bacteria also exhibit antagonistic interactions in communities, i.e., benefiting from cell separation (Russel et al., 2017; Peterson et al., 2020). Such antagonistic effects include the competition for limited resources, in particular among metabolically similar species as encountered in

the atmosphere. The functioning and survival of bacterial communities is usually due to a balance between mutualism and antagonism. However, the specific conditions facilitating such balance in different ecosystems may shift under atmospheric conditions, and therefore influencing bacterial metabolism and survival. Reduced antagonism in the atmosphere due to the physical separation of cells through aerosolization, may partly explain the sustained activity and survival of atmospheric bacteria despite the overall harsh environmental conditions.

The role of such substantial distances between cells in the atmosphere has not been explored yet as lab experiments are usually performed on bulk samples. For example, Vaïtilingom et al. (2010, 2011) derived biodegradation rates in cloud water samples ($\sim$100 mL) that combined billions of droplets containing 10,000s of cells. Such bulk experiments are convenient and often the only way to monitor signals above the detection limit, but they do not reflect the same conditions that microorganisms experience in dispersed cloud droplets (Sections 2.3 and 2.4.2). In addition, the chemical microstructure in individual droplets

may be different than in bulk solutions (Wei et al., 2018). Novel experimental set-ups that allow the investigation on levitated droplets containing microorganisms seem promising to overcome current methodological limitations (Fernandez et al., 2019).

## 2.2 Water availability: are atmospheric microbial oases limited to clouds?

All organisms, including microorganisms, need water for their biological functioning. The water content of surface waters is unlimited (at least - presumably - from a microbial perspective); the water content of soils is typically on the order of several

volume percent. Clouds represent air masses with the largest amounts of liquid water in the atmosphere, but yet they only comprise $10^{-5}$ - $10^{-4}$ volume% of the total atmosphere, corresponding to typical liquid water contents of $0.1 < \text{LWC}_{\text{cloud}} < 1$ g m$^{-3}$. Even in the absence of clouds, aerosol particles are never completely dry. In fact, aerosol water is often a substantial fraction of a particle mass depending on particle hygroscopicity and the surrounding relative humidity (RH). The total aerosol water content might amount to several 100s $\mu\text{g}_{\text{H2O}}$ m$_{\text{air}}^{-3}$, corresponding to $\sim10^{-8}$ volume%.

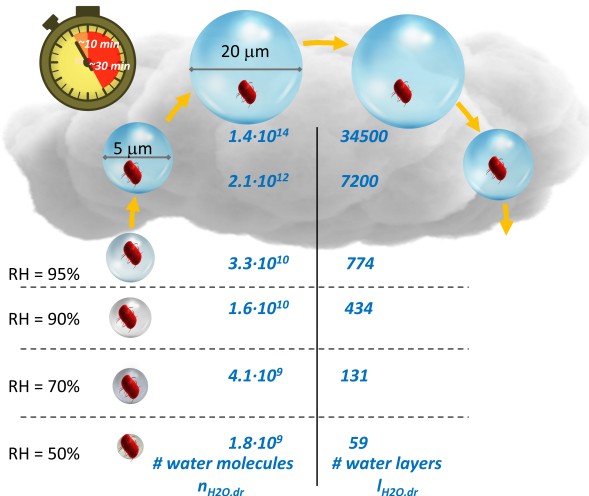

**Figure 3.** Number of water molecules ($n_{H2O,dr}$) and water layers ($l_{H2O,dr}$) under subsaturated conditions (RH < 100%) and for cloud droplets with diameters of 5 $\mu$m and 20 $\mu$m (Section S2.1, supplemental information). The schematic suggests that under a wide range of atmospheric RH conditions, bacteria cells are surrounded by at least $\sim$60 water layers which corresponds to about 20 mass% of the water in a bacteria cell ($m_{cell,H2O} \sim 1.3 \cdot 10^{10}$ molecules water per cell). Since typical lifetimes of cloud droplets are on the order of 10 - 30 minutes, the largest water volumes may only represent very short-lived 'oases' (Section S2.2, supplemental information).

The water content of atmospheric aerosol particles is often expressed by means of the hygroscopicity parameter $\kappa$. Strictly, the concept of hygroscopicity does not apply to bacteria cells as they do not fully dissolve in water; their water uptake may be triggered by the formation of biosurfactants that partition to the air-water interface that attract water and slow down water evaporation to trap the water in the immediate surroundings of the cell (Gill et al., 1983).

  Using $\kappa_{bact}$ = 0.1 to estimate the amount of water around a cell, the number and layers of water molecules, $n_{H2O}$ and

$l_{H2O}$, can be derived (Figure 3 and Section S2.1 in the supplemental information): at RH $\sim$90%, the water masses inside and

outside of a cell are approximately equal ($\sim10^{10}$ water molecules). At RH $\sim$50%, there are several tens of layers of water around a bacteria cell ($l_{H2O} = 59$), corresponding to about 20% of the water inside a bacteria cell. Figure 3 summarizes $n_{H2O}$ and $l_{H2O}$ values over the range of RH or water activities ($a_w \sim$ RH/100%), in which microbial activity, including cell division, has been observed (Stevenson et al., 2015, 2017). The presented $n_{H2O}$ and $l_{H2O}$ values are possibly biased low as they imply hygroscopic growth that often occurs only immediately after particle emission in the atmosphere. After their first deliquescence event, particles usually stay in a metastable state above their efflorescence RH which implies that they retain condensed water due to hysteresis of evaporation. In addition, during their residence time in the atmosphere, hygroscopic material likely condenses on bacteria-containing particles, resulting in even more water uptake. Moreover, Nielsen et al. (2024) showed that electrolytes associated with the cell, such as sodium chloride, may significantly increase the water uptake.

When RH exceeds 100%, cloud droplets may form, depending on the supersaturation that is a function of the cooling rate (vertical velocity) as a source and the available drop surface area as a condensational sink. While clouds can exist for several hours or even days, they are highly dynamic systems. Cloud droplets form near cloud base and grow while ascending, followed by shrinking in descending (i.e., warming) air masses (Figure 3). These up- and downward motions constrain the lifetime of an individual cloud droplet to about 10 - 30 min, depending on cloud thickness and vertical velocity (Section S2.2 in the Supplemental Information). Thus, such high-water-volume conditions may only exist for very short times during which bacteria need to adapt to the in-cloud environment that greatly differs from that outside clouds in terms of water availability, light, temperature etc. Lab experiments in (artificial) cloud water revealed lag times of several hours before bacteria started to efficiently biodegrade organic cloud water constituents (Vaïtilingom et al., 2013). This lag time is comparable to that as observed for bacteria growth after rewetting of soil ('Birch effect') (Leizeaga et al., 2022). If indeed conditions of high bacterial activity were limited to the in-cloud time, hysteresis effects might impede enhanced microbial activity inside clouds since droplets evaporated before the end of the lag period.

Clouds have been previously termed 'microbial oases' in the atmosphere (Amato et al., 2017), implying that they provide the most ideal atmospheric conditions for bacterial activity in a fugace habitat. However, given the very rapid changes over short time scales in the environmental conditions, clouds may actually trigger several stress responses that differ from those under lower RH (20 - 90%) conditions (Péguilhan et al., 2023, 2024). Provided that bacteria are surrounded by a considerable amount of water at $\gtrsim$ 90% RH, the range of biologically favorable conditions might actually extend to much longer temporal and spatial scales than estimated previously (Ervens and Amato, 2020). Such considerations might be relevant for regions and/or periods of high relative humidity (e.g., tropics, polar regions and/or during night-time). It remains to be explored whether microorganisms in the atmosphere adjust their stress responses and biological functions over the full continuum of relative humidity (RH) conditions, rather than switching behaviors at specific RH thresholds (e.g., inside/outside clouds).

## 2.3 Accessibility to nutrients: are nutrient levels in droplets high enough?

In the atmosphere, bacteria cells may exhibit different levels of metabolic activity, which range from mere survival strategies, i.e., activity focused solely on repairing cellular damage, to dormancy, during which cells sustain their essential biological functions, to growth and multiplication as the most energy-intensive activities (Price and Sowers, 2004). Cells may become

dormant under water-limited conditions (Haddrell and Thomas, 2017) or due to other stressors (Šantl-Temkiv et al., 2022). In cloud water, Sattler et al. (2001) observed cell activity at $0°C$ compatible with cell growth, whereas dormancy was observed outside clouds (Smets et al., 2016). Given that particles (including bacteria calls) only spend a fraction of their time inside clouds (Ervens and Amato, 2020), it can be, thus, expected that many bacteria may be dormant for long periods of their atmospheric residence time. Dormancy has been shown in other environments to be an efficient response to harsh conditions

and ultimately being beneficial for survival (Jones and Lennon, 2010).

    In most natural waters the concentrations of organic nutrients are sufficiently high that they do not lead to carbon-limiting conditions for heterotrophs (Eiler et al., 2003). In oligotrophic ocean regions or lakes, bacteria develop strategies to adapt to the lower nutrient levels, including the optimization of their energy use by producing less bacterial secondary mass, while enhancing respiration (del Giorgio and Cole, 1998). The main organic nutrients for bacteria in rivers are low-molecular-weight

compounds (LMW with $\lesssim$ 500 Da) that comprise a major fraction of the total dissolved organic carbon (TOC) (Catalán et al., 2017). The total concentration of biodegradable LMW compounds in surface waters is on the order of 50 $\mu$mol L$^{-1}$ (0.005 - 0.4 $\mu$mol L$^{-1}$ for individual compounds; Table S3), which overlaps with the range as found for individual compounds in fog and cloud water (0.1 - 10 $\mu$mol L$^{-1}$, Herckes et al. (2013)). This similarity is not surprising as volatile, water-soluble compounds are continuously emitted and/or formed in the connected spheres, followed by (thermodynamic) partitioning at the

atmosphere/water interfaces of cloud droplets and surface waters. The incorporation of bacteria cells into the aqueous phase does not follow the same thermodynamic principles, but, yet, bacteria cell concentrations in river and sea water ($\sim10^3$ - $10^5$ cell mL$^{-1}$), are comparable to those in cloud water ($10^4$ - $10^6$ cell mL$^{-1}$) (Amato et al., 2017).

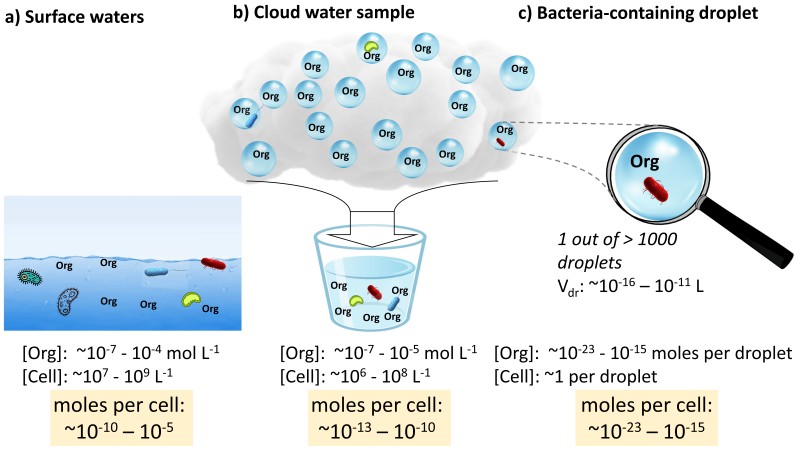

**Figure 4.** Typical concentrations of bacteria cells and organics in surface waters (Table S3) and cloud water samples (Herckes et al., 2013) and resulting organic-to-cell ratios. Given that 1 out of > 1000 cloud droplets contains a bacteria cell, this bulk ratio is reduced by several orders of magnitude in individual bacteria-containing cloud droplets (< 10$^{-23}$ - 10$^{-15}$ moles per cell; Section S3 in the supplemental information), potentially suggesting limited availability of organic nutrients in cloud droplets.

The similarity in organic nutrient levels and bacteria cell concentrations in the different water phases may suggest equivalent organic-to-cell ratios. However, this conclusion only applies to the comparison of bulk aqueous volumes, e.g., cloud water samples where all bacteria-containing and bacteria-free cloud droplets are combined (Figure 4b). In contrast, within the microcosms of individual bacteria-containing cloud droplets, the organic-to-cell ratio is much smaller since the cell concentration is 1 per droplet (0 in > 99.9% of the droplets) while the solute concentration [mol L$^{-1}$] is that of the bulk cloud water sample. This results in organic-to-cell ratios in cloud droplets of < 10$^{-15}$ moles per cell as opposed to a range of 10$^{-13}$ to 10$^{-10}$ in surface waters (Figure 4c and Section S3). As a consequence, the steady-state nutrient levels per cell in the bacteria-containing droplets are at least 2, possibly up to 10, orders of magnitude lower than those in other aquatic environments.

Previous lab experiments to derive biodegradation rates in cloud water used organic-to-cell ratios in the range of 10$^{-12}$ - 10$^{-10}$ moles cell$^{-1}$ (Vaïtilingom et al., 2010, 2011). Process models explored the role of biodegradation on chemical budgets in the atmospheric multiphase system (Khaled et al., 2021; Nuñez López et al., 2024) and found that the nutrient uptake from the gas phase may not be sufficiently fast to replenish biodegraded organics in the small subset of droplets, in which microorganisms are present and potentially active, resulting in even lower nutrient-to-cell ratios. In the same studies, it was shown that simplified model assumptions of an 'averaged cell concentration' in each droplet (i.e., < 0.001 cell droplet$^{-1}$, e.g., Pailler et al. (2023)) may lead to wrong conclusions regarding the role of biodegradation, as the nutrient-to-cell ratio is substantially overestimated. In light of these considerations, we suggest that lab and model studies should be performed for droplet-relevant organic-to-cell ratios to systematically explore potential limitation thresholds for individual bacteria strains and nutrients. Results from such studies would lead to a more accurate categorization of the atmosphere in terms of the trophic level. They could be also used to support (or refute) the classification of bacteria in rain as 'extremophiles' as put forward by Guillemette et al. (2023) based on much lower TOC levels in rain as compared to marine surface waters and in the deep sea.

Extrapolating the conclusions as made above for clouds to potential biological activity in aerosol particles (diameter $\lesssim$ 1 $\mu$m) possibly suggests even greater nutrient limitation at RH < 100%. However, the partitioning of LMW volatile organics into aerosol water can greatly differ from that to (relatively) dilute droplets. For example, the fraction of formic acid partitioned to aerosol water is up to 7 orders of magnitude higher than that predicted based on Henry's law (Liu et al., 2012). Such strong partitioning could compensate for the lower water volume. Thus, one may hypothesize that nutrient-to-cell ratios in wet aerosol particles and cloud droplets are similar. However, since the water and oxidant contents are different in aerosol particles and cloud droplets (Sections 2.2 and 2.4.2), the resulting osmotic and oxidative stress levels in these two different aqueous regimes might lead to different nutrient utilization rates.

## 2.4 Sunlight and oxidants: is the high photochemical activity in the atmosphere always stressful?

### 2.4.1 Solar radiation

The photic zone of surface waters, i.e., the layer penetrated by sunlight, is limited to the first few meters below the air-water interface; in soil, this layer is even shallower and restricted to a few milli- to centimeters. In all environments, including the atmosphere, the photolysis of dissolved organic matter results in LMW compounds that are often more biodegradable. At the

same time, products of photochemical reactions include reactive oxygen species (ROS), such as the OH radical, that oxidize organics reducing the concentration levels of organic nutrients (Scully et al., 2003). In addition, ROS cause oxidative stress to bacteria cells as they can damage the cellular structure. Therefore, sunlight has both beneficial and adverse effects on microbial functioning and survival. Airborne particles are exposed to sunlight about half of their atmospheric residence time (during the day), as opposed to the much lower photochemical activity in surface environments. DNA damage due to UV light in the atmosphere can be parameterized as a function of irradiance and bacteria type (Madronich et al., 2018).

The actinic flux inside clouds represents a particular challenge as it can be enhanced or reduced as compared to cloud-free air depending on cloud optical density and the height in cloud (Ryu et al., 2017). Enhanced actinic fluxes occur when light is scattered or reflected multiple times inside 'bright' clouds with low optical thickness and few and/or small droplets. Conversely, sunlight cannot readily penetrate through dense, 'dark' cloud, resulting possibly in lower UV-induced stress for bacteria. Thus, the macro- and microstructures of clouds (e.g., vertical profiles of density) have to be taken into account to estimate the time scales during which bacteria are exposed to detrimental UV conditions. To counteract such UV-induced stress, bacteria may develop strategies mechanisms such as the formation of pigments as a protective shield. However, such mechanisms may only become effective after some lag time that has not been quantified under atmospheric conditions.

### 2.4.2 OH radical

The OH radical is one of the most powerful oxidants and 'cleansing agent' in the atmosphere and aquatic environments. Its main formation pathways in surface water include the direct photolysis of hydrogen peroxide ($H_2O_2$) and the Fenton reaction (Fe(II) oxidation by $H_2O_2$). In the atmosphere, it is mainly formed in the gas phase via the photolysis of ozone and HONO but also by the reaction of the hydroxy peroxy radical ($HO_2$) with NO. The rate of the latter process is significantly reduced in cloudy air masses, since the highly soluble $HO_2$ partitions into cloud water where it is quickly consumed, while the less soluble NO remains in the gas phase. This reactant separation leads to significantly lower total $HO_2$ and OH concentrations in the presence of clouds (reductions of up to 70% and 80%, respectively, Ervens (2015)).

At typical cloud liquid water contents, less than 0.1% of all atmospheric OH radicals reside in cloud water. The aqueous phase concentration in cloud water [mol L$^{-1}$] is greater by about 3 orders of magnitude than in sea water (Figure 5a *vs.* b). This comparison may imply - on a first sight - higher oxidative stress due to OH in the atmosphere. However, this conclusion may need to be carefully reconsidered if one defines the number of OH radicals in the immediate surrounding of a bacteria cell as a direct measure of the OH-oxidative stress for bacteria. Figure 5c shows exemplary that the steady-state OH concentration per droplet only exceeds 1 in droplets with diameters $D_{dr} > 20$ $\mu m$ (Section S4 in the supplemental information). It should be kept in mind, though, that steady-state concentrations are a result of OH source and loss rates that usually (nearly) cancel. OH sources in cloud droplets include the direct uptake from the gas phase and chemical reactions with rates of ∼10 - 10000 OH radicals per second (Figure 5c), with the Fenton being one of the main sources (Figure S6b). However, unlike the other aqueous phase OH sources (e.g. $H_2O_2$ or $NO_3^-$ photolysis), the Fenton reaction only occurs in the subset of cloud droplets that contain iron (∼10%, Khaled et al. (2022); Ervens (2022)). Thus, the oxidative stress due to Fenton chemistry as observed in other aquatic environments (Cabiscol et al., 2000), may be much lower in the atmosphere and only occur in 0.01% of all

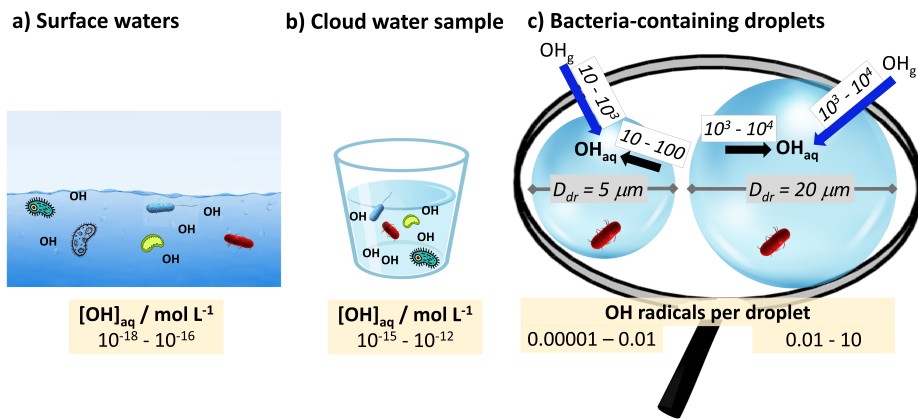

**Figure 5.** Concentrations of the OH radical in a) sea water (Mopper and Zhou, 1990); b) cloud water (Arakaki et al., 2013); c) individual cloud droplets ($D_{dr} = 5\mu$m, $20\mu$m). The blue arrows in panel c denote the uptake rates of OH radicals per second, the black arrows denote their production rate inside a droplet (cf Figure S6c for individual reactions).

droplets or particles (= $10\% \times 0.1\%$ assuming a statistical distribution of iron and bacteria across a drop population). The OH uptake from the gas phase is often not sufficiently fast to replenish the consumed radicals and reach equilibrium concentrations ($\sim 10^{-12}$ mol L$^{-1}$), resulting in the lowest OH concentrations in large droplets (Ervens et al., 2014).

Microorganisms apply strategies to scavenge oxidants and respond to oxidative stress. The conditions under which oxidant scavenging rates cannot be compensated by defense mechanisms depend on numerous factors, including the bacteria species, oxidant (concentration and species) and environmental conditions, such as temperature, relative humidity and nutrient availability. Systematic studies to constrain such conditions have not been performed yet in dispersed droplets. The aqueous photooxidation experiments of living bacteria in the presence of OH by Liu et al. (2023) might serve as a first indication; however, the observed maximum survival time of 6 hours in a large aqueous phase volume likely represents a lower limit as compared to realistic in-cloud processing times during which OH-induced stress levels may be lower.

### 2.4.3 Other ROS and oxidants (e.g., $^3$C*, $^1$O$_2$, H$_2$O$_2$)

While the OH radical is one of the most powerful chemical oxidants in the atmosphere, other oxidants are present at high concentrations as well, including $H_2O_2$, $HO_2$, triplet states ($^3$C*), singlet oxygen ($^1O_2$) and ROS formed by photosensitizers which all cumulatively contribute to the oxidative stress of bacteria in clouds. Due to its high total concentration and solubility, $H_2O_2$ represents (one of) the most abundant ROS in cloud droplets. Wirgot et al. (2019) showed that microorganisms can metabolize $H_2O_2$ in aqueous solution with a chemical composition similar to that of cloud water. The experiments were conducted in a closed system; thus, it was not assessed whether biodegradation was sufficiently fast to completely detoxify a droplet that is continuously exposed to a gas phase $H_2O_2$ reservoir. Simultaneous measurements of $H_2O_2$ in the atmospheric

gas and aqueous phases usually show thermodynamic equilibrium. However, such measurements reflect the conditions in the bulk cloud water; they do not allow the detection of small deviations due to biodegradation in a few droplets.

The concentrations of the more reactive $^3C^*$ and $^1O_2$ in cloud droplets are 1 to 2 orders of magnitude higher than that of OH (Kaur and Anastasio, 2018). Thus, scaling up the results shown for the OH radical in Figure S6a by a factor 10 - 100 suggests that there are a few $^3C^*$ and $^1O_2$ in each cloud droplet. The formation rates of these oxidants under typical cloud conditions have not been fully characterized and implemented in multiphase chemistry models, and their metabolic consumption are even less constrained. Thus, the extent cannot be assessed to which the particular oxidant mix in cloud water - that is likely different

from that in other aquatic environments - affects oxidative stress levels and ultimately, metabolic functioning and survival in the atmosphere. As compared to cloud droplets, the concentrations of $^3C^*$, $^1O_2$ and $H_2O_2$ in aerosol water are higher by 1 to 2 orders of magnitude (Arellanes et al., 2006; Ma et al., 2023) whereas OH concentrations are similar in both systems. However, the chemical conversion rates of the oxidants might be much higher than in droplets due to higher precursor concentrations (e.g. $NO_3^-$). At the same time, the microorganisms are exposed to higher ionic strength in the much less dilute particles that

might alter their ability to apply efficient antioxidative defense mechanisms. Such mechanisms could include the formation of biosurfactants that slow down or even impede uptake from or evaporation to the gas phase (Gill et al., 1983). Due to the lack of kinetic and mechanistic data to describe chemical formation and loss rates of ROS species in aerosol particles, the ROS budgets in aerosol water cannot be speciated on a molecular level. Therefore, dedicated lab experiments and process model studies should be designed to constrain oxidant levels in the atmospheric aqueous phases to ultimately determine their effect

on microbial oxidative stress levels.

### 2.4.4 pH value

Numerous studies point to the fact that atmospherically relevant bacteria (e.g., *Pseudomonas sp.*) show highest growth and activity rates at only mildly acidic or neutral pH values. However, they have developed various strategies to survive pH conditions outside these ranges. They include intracellular buffering to maintain the pH within the cytoplasm or proton pumps that

regulate the intracellular proton concentrations (Lund et al., 2020; Kobayashi et al., 2000). Such mechanisms may explain the weak dependence of biodegradation rates on pH as found in lab studies of (artificial) cloud water (Vaïtilingom et al., 2010) as the intracellular pH is kept at (near) neutral values. Liu et al. (2023) found different trends when they examined the pH dependence of the survival and biodegradation rates of two strains of *Enterobacter* bacteria isolated from ambient air in a polluted environment: They showed that in the presence of light, the survival rate decreased in particular at pH $\leq$ 5. These

trends may point to different sensitivities of this particular bacteria type to pH, as compared to the responses by bacteria in cloud water (Vaïtilingom et al., 2013). The concurrent responses to low pH and the presence of sunlight may suggest some photolytic or photochemical mechanism that influences the biodegradation activity. In the limited volume of a cloud droplet or aqueous aerosol particle, the number of protons is small as compared to that in bulk aqueous phases. Thus, the individual cells in such volumes have to 'combat' a limited number of protons (Figure S7 in the supplemental information). Thus, the

adjustment of the pH of the surrounding aqueous phase as observed in other environments (Ratzke and Gore, 2018), might be

easier in the small droplet or particle volumes. Such rates of buffering agent production and proton transfer likely depend on the bacteria types and environmental factors and therefore should be explored for atmospheric conditions.

# 3 Trends of atmospheric microorganisms and pollutants: correlation, causation or coincidence?

Observations of atmospheric trends between chemical concentrations and bacterial abundance and community structure (including diversity) are to a large extent determined by air mass types, history and age (Gandolfi et al., 2013; Innocente et al., 2017). Several atmospheric studies aimed at identifying atmospheric chemical or physical parameters that influence the abundance, diversity and/or viability of atmospheric microorganisms. Such studies often resulted in contradictory conclusions regarding the impact of specific atmospheric conditions (e.g., pollution or clouds) on the atmospheric microbiome. Lebowitz and O'rourke (1991) cautioned against concluding on links between aerobiological and chemico-physical contaminants in determining factors that potentially trigger adverse health effects. In view of the microscale chemical and microphysical factors discussed in Section 2, similarly we emphasize the importance of not confusing causation with correlation regarding atmospheric parameters affecting microbial stress.

Figure 6 summarizes chemical or microphysical factors in polluted air masses that may affect different types of microbial stress levels (oxidative, osmotic, UV induced, transitional). None of the shown effects imply a direct adverse or beneficial influence by a specific pollutant (e.g., $SO_2$, $NO_x$ VOCs or their oxidation products) on bacteria activity level, viability, diversity or abundance. We do not suggest that such effects do not exist; however, to the best of our knowledge, corresponding biomarkers (specific response genes, enzymes) have not been identified yet. Also, we do not propose that indeed all listed chemical or microphysical parameters trigger significant stress responses or that any of the indicated responses dominate or cancel each other. Instead, the schematic may be used as a qualitative - or even speculative - illustration to motivate studies quantifying the potential role of the individual factors in targeted experiments. The results of such studies may then be used to conclude on the role of each of these factors to derive robust cause-and-effect relationships. Falsely derived conclusion of causation may ultimately result in inappropriate strategies to maintain healthy ecosystems and biodiversity in a changing atmosphere.

In cloud-free air masses, an increase in the hygroscopicity of the bacteria-containing particle may lead to lower osmotic stress, but possibly to more efficient ROS uptake from the gas phase (Figure 6a). Whether such a protection mechanism is significant at all or negligible as compared to the lower OH levels that exist in polluted air masses remains to be seen.

As pointed out in the previous sections, many microphysical processes determine the micro- and macroscopic structure of clouds, including droplet size distributions that determine droplet lifetime (smaller droplets are less likely to precipitate) and cloud optical thickness (Section 2.4.1), which may have opposite effects on stress levels (Figure 6b). In turn, higher particle concentrations in polluted air masses result in smaller, more numerous droplets with higher solute and OH concentrations, i.e., enhancing osmotic and oxidative stress. However, since OH (and several other oxidant) levels are reduced in the presence of clouds (Section 2.4.2), an apparent lower stress level in the presence of clouds could be possibly just ascribed to exposure to lower oxidant levels.

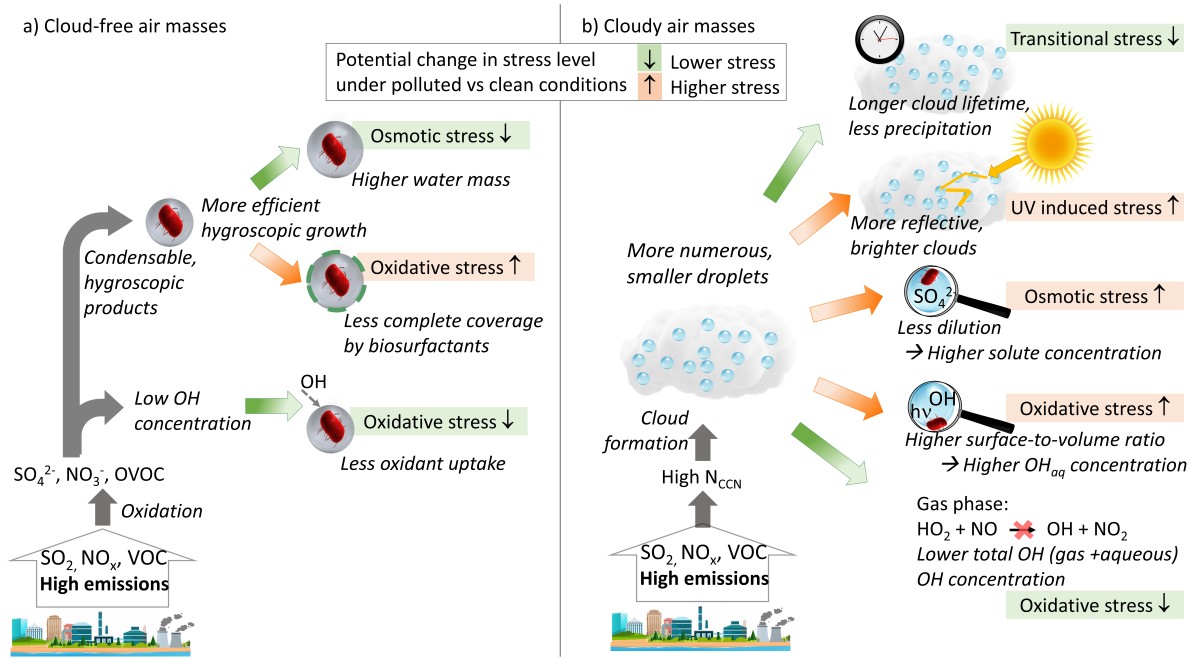

**Figure 6.** Potential microbial stress responses due to chemical or microphysical processes in polluted air masses. It is not implied that each of these factors is necessarily important or significant to have beneficial or adverse effects on bacteria. Instead, the illustration intends to demonstrate that correlations between atmospheric pollutants do not imply causation. a) Cloud-free air masses: osmotic stress may be lowered due to addition of hygroscopic mass that attracts more water (Section 2.2) whereas oxidative stress may increase if particles exceed sizes too large to be covered by biosurfactants to shield the particles from oxidant uptake. Total atmospheric oxidant levels may be comparably small in a polluted atmosphere since OH is efficiently titrated (Section 2.4.2). b) Cloudy air masses: high aerosol number concentrations lead to clouds with more numerous but smaller cloud droplets ('cloud lifetime effect'). Such clouds are less likely to precipitate but are, at the same time, less dense and brighter as light reflections between droplets are amplified (Section 2.4.1). Smaller droplets typically contain higher solute and oxidant (OH) concentrations which may enhance both osmotic and oxidative stressors. The latter may be (partially) compensated for by the fact that total (gas + aqueous) OH and $HO_2$ concentrations are usually much lower in the presence of clouds due to smaller formation rates in the gas phase (Section 2.4.2).

Such considerations are likely just a small subset of bio-physico-chemical feedbacks in the atmospheric multiphase system to explain observed trends. Adding to the vagueness is the lack of comprehensive, speciated bacterial emission maps and patterns. Bacteria are detached from the surface and lifted by mechanical forces (e.g., strong wind), just like any other primary particle. In addition, there might be biological selection criteria that trigger aerosolization and emission of specific microorganisms. Since bacteria emission fluxes into the atmosphere are poorly constrained, observed trends between bacteria-related parameters and other environmental factors may be just coincidental since the atmospheric composition is a result of mixing of air masses of different origin and age. 'Chemical clocks', e.g., the ratio of co-emitted but differently reactive compounds, are often used to

determine the 'chemical age of air'. Equivalent measures to determine similar indicators for the 'biological age of air' are still lacking. Such indicators would be useful to not only determine the viability and survival along air mass trajectories but it would also allow to conclude on emission patterns at different locations.

## 4    Conclusions

Atmospheric microbiology is an interdisciplinary research field at the intersections of atmospheric sciences, aerobiology and
microbial ecology. Figure 7 illustrates their overlap and lists some shared research objectives (biogeochemical cycles, microbial diversity and dispersion, and atmospheric transport, respectively). Scientific concepts of all three areas have to be taken into account for a comprehensive characterization of atmospheric microorganisms as a small, but yet important, dynamically rapidly changing and evolving fraction of Earth' microbiome.

      Numerous previous studies focused on the role of microorganisms on the atmosphere for impacting clouds or chemical
concentrations. By now, there is general consensus that this role may be limited to specific regions, conditions and processes such as ice nucleation near 0°C, or biodegradation of selected compounds as a significant atmospheric sink. However, despite the limited global influence of microorganisms on the atmosphere and climate, further studies of the microbial bio-physico-chemical properties are needed to explore the ability of microorganisms to take up water and/or be incorporated in clouds (hygroscopicity, ice nucleation ability). This information will be important to correctly estimate the atmospheric processing
times of microorganisms and, therefore, their fate. Such studies can apply the same methodologies used for other atmospheric particles, but should be motivated by research questions targeted to understand the role of the atmosphere on microorganisms, rather than the reverse. Similarly, the rationale for exploring biodegradation rates in cloud water could be extended from focusing on potential impacts on chemical budgets to consequences of limited nutrient availability on levels of metabolic activity, including dormancy, starvation and survival.

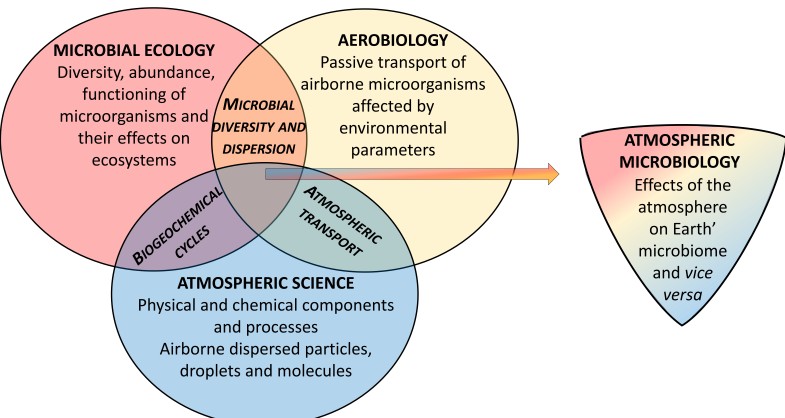

**Figure 7.** Atmospheric microbiology forms the intersection of atmospheric sciences, microbial ecology and aerobiology

Studies on microbial optimization strategies for water uptake or nutrient utilization are not new; they have been conducted for many decades in the context of aquatic and terrestrial environments. Transferring such established methods to the atmospheric microbiome is challenging for (at least) two reasons: Firstly, atmospheric bacteria number concentrations are extremely low. Thus, collecting statistically meaningful samples requires large volumes and/or long time scales. This complicates the identification of patterns and sources as air masses efficiently mix over time. Despite the difficulties associated with the statistically

relevant sampling of atmospheric microorganisms, individual environmental factors have been started to be identified that control the concentration (Archer et al., 2019; Gusareva et al., 2019), diversity (Tong and Lighthart, 1997; Bryan et al., 2019) and selection (Smith et al., 2011; Joly et al., 2015) of atmospheric microorganisms. While it remains difficult to perform sampling with sufficiently high resolution and frequency, dedicated strategies should be developed to constrain the role of individual environmental parameters (e.g., UV light, temperature) for microbial diversity and survival in the atmosphere. Such studies

should be accompanied by suitable lab or chamber studies under controlled conditions to test hypotheses that may be formed based on ambient observations.

Secondly, unlike in other environments where bacteria can form aggregates and biofilms or are suspended in water, atmospheric bacteria are aerosolized before they are lifted and dispersed in air. The immediate environment of airborne bacteria is constrained by particle or drop volumes. We highlight such specific conditions due to bacteria aerosolization, that imply

distances of several centimeters between cells, which makes their living environments different from those in (more) homogeneous environments. Previously, conclusions on potential stress factors or nutrient availability in the atmosphere were drawn based on the concentration levels in atmospheric bulk samples where $\lesssim 0.1\%$ bacteria-containing droplets are mixed with $\gtrsim 99.9\%$ bacteria-free droplets (particles). In dispersed droplets (particles), the concentration ratios of cells to nutrients or oxidants, respectively, should be considered on a per-droplet (per-particle) basis. The much lower cell-to-nutrient ratio in

an individual droplet ($\lesssim 1/0.1\%$) as compared to that in bulk samples implies a much lower trophic regime for bacteria than previously thought. The atmosphere is considered a harsh environment for microorganisms due to numerous factors, including high photochemical activity. However, droplets and particles contain only a few oxidant molecules (e.g., OH, $^3C^*$, $^1O_2$) that are quickly transformed chemically but possibly also metabolized. To describe the particular mixtures of nutrients and oxidants that bacteria are exposed to in the aerosol/droplet microcosms, process models should be applied with detailed atmospheric mul-

tiphase chemistry, microphysics and data on biological processes (e.g., biotransformation of nutrients or ROS). However, this latter data is largely lacking, in particular from studies on individual (e.g., levitated) droplets or particles. Given the challenges associated with such experimental set-ups, we recommend that bulk experiments should at least take into account realistic solute mixtures.

In summary, we provide a new perspective on atmospheric microbiology within the context of atmospheric chemistry and

microphysics. We emphasize the importance of specific spatial and temporal scales of microbial microcosms in individual particles and droplets. Such considerations are essential to enhance our understanding of the atmosphere as an extension of the more well-characterized microbial environments such as oceans and soil. Ultimately, this will lead to a complete characterization of the Earth microbiome and its cycling that ensures global ecological stability and functioning.

*Code and data availability.* The supplement contains all equations that were applied for calculations using the cited literature values.

*Author contributions.* BE developed the idea and concept of the study. All co-authors contributed to discussions and to the writing of the manuscript.

*Competing interests.* The authors do not have any competing interests to declare.

*Acknowledgements.* We thank Z. Bourhane, A.-M. Delort, F. Rossi and J. Vyskocil for useful discussions.

*Financial support.* This work has been supported by the French National Research Agency (ANR) (grant no. ANR-17-MPGA- 0013).

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
