# Peer review of "Ideas and perspectives: Microorganisms in the air through the lenses of atmospheric chemistry and microphysics"

_EGUsphere, 2024_

## Author Comment (AC1)

**Author response to comments by Referee #1**
All referee comments are shown in black, our author responses in blue; suggested new manuscript text is indicated in red.

This perspective was an interesting and enjoyable to read contribution that identifies and more closely examines some of the most salient challenges that microbes in the atmosphere encounter. A deeper examination of these challenges reveals that some of them may be as or more challenging that previously believed, while some may be less relevant, due to the unique circumstances of the atmosphere, and in cloud and aerosol phases. The conclusions presented are mainly based on modelling, but based on realistic approximations and our currently knowledge. As yet, little empirical data exists to validate many of these suppositions. However, the authors seem to be cognizant of these limitations (in referencing literature where unexpected findings were made e.g. Liu et al. 2012). Nevertheless, the perspective builds a more detailed and nuanced examination of these factors than I have seen elsewhere.
**Author response**: We thank the referee for their very positive assessment and the constructive suggestions. We address the comments below.

Specific comments:

1) While microbial interactions can have beneficial outcomes, there can also be negative outcomes/ antagonistic interactions. The physical separation of cells that is more prevalent in the atmosphere than in other environments will not only reduce/eliminate beneficial interactions, but will reduce/ eliminate negative interactions such as competition and direct antagonism. The net outcome on this may be beneficial or detrimental, which likely depends on the impacted organism and the context.
**Author response**: We thank the referee for raising this aspect. To the best of our knowledge, there are no specific studies that explored the benefits of the interbacterial antagonism in the atmosphere. Therefore, we add some general discussion of the concept of antagonism at the end of Section 2.1:
In addition to such mutualistic behavior, bacteria also exhibit antagonistic interactions in communities, i.e., benefiting from cell separation (Russel et al., 2017; Peterson et al., 2020). Such antagonistic effects include the competition for limited resources, in particular among metabolically similar species as encountered in the atmosphere. The functioning and survival of bacterial communities is usually due to a balance between mutualism and antagonism. However,the specific conditions facilitating such balance in different ecosystems may shift under atmospheric conditions, and therefore influencing bacterial metabolism and survival. Reduced antagonism in the atmosphere due to the physical separation of cells through aerosolization, may partly explain the sustained activity and survival of atmospheric bacteria despite the overall harsh environmental conditions.

2) I appreciated the discussion on water activity, and the calculations estimating the potential hydration shell that might exist under different conditions. One aspect that was missing for me in this perspective is the impact of pH in aerosols and clouds on microbial cells – most microbes are not well adapted to living at pH 5, so this presents another stressor for microbes in the atmosphere.
**Author response**: Firstly, regarding the water uptake, we would like to note that we became aware of a new paper by Nielsen et al. (2024) that adds valuable information on water uptake by *Pseudomonas sp.* We will refer to it in Section 2.1 and in the supplemental information.
Secondly, we agree with the referee that the pH is a very important parameter that affects microbial activity and survival. Numerous studies point to the fact that environmentally relevant bacteria show highest diversity and richness near neutral pH values (Fierer and Jackson, 2006). However, bacteria developed numerous strategies to survive other pH conditions outside this pH range. They include intracellular buffering to maintain the pH within the cytoplasm (Kobayashi et al., 2000) and adjustments of the proton concentrations in their surroundings (Ratzke and Gore, 2018). Such buffering may also explain the low dependence of biodegradation rates in cloud water on pH (within a range of 3 - 6) as observed in lab studies, since such processes occur inside the cell even though the extracellular pH may vary over several orders of magnitude (Vaïtilingom et al., 2010; Nuñez López, 2024). In line of the considerations in the present paper, we add the following text as a new subsection in section 2.4:
**pH value:** Indeed, numerous studies point to the fact that atmospherically relevant bacteria (e.g., *Pseudomonas sp.*) show highest growth and activity rates at only mildly acidic or neutral pH values. However, they have developed numerous strategies to survive other pH conditions outside these ranges. They include intracellular buffering to maintain the pH within the cytoplasm or proton pumps that regulate the intracellular proton concentrations (Lund et al., 2020; Kobayashi et al., 2000). Such mechanisms may explain the weak dependence of biodegradation rates on pH as found in lab studies on (artificial) cloud water (Vaïtilingom et al., 2010) as the intracellular pH is kept at (near) neutral values.

In the limited volume of a cloud droplet or aqueous aerosol particle, the number of protons is small as compared to that in bulk aqueous phases. Thus, the individual cells in such volumes have to 'combat' a limited number of protons (supplemental figure). Thus, the adjustment of the pH of the surrounding aqueous phase as observed in other environments (Ratzke and Gore, 2018), might be easier in the small droplet or particle volumes. Such rates of buffering agent production and proton transfer likely depend on the bacteria types and environmental factors and therefore should be explored for atmospheric conditions.

[Figure]

Figure 1: Number of protons in the aqueous volume of a particle or droplet surrounding a bacteria cell with a diameter of 1 $\mu$m

3) Several studies are beginning to build convincing links between environmental factors (e.g. UV, temperature) on microbial community structure and diversity in the aerosol phase (e.g. Archer et al. 2019 Nature Microbiology, Gusareva et al. 2019 PNAS). Developing a controlled experimental system for forming these links is difficult to conceptualize, but environmental sampling is challenged by the numerous confounding variables. With sufficiently well-resolved and controlled environmental sampling, disentangling the impacts of specific variables should be feasible, for at least some environmental factors.

**Author response**: We agree with the referee that controlled environmentally sampling with a statistical meaningful number of samples may be a promising strategy to constrain the role of individual environmental factors. Such sampling strategies can and should be accompanied by suitable lab or chamber experiments to test hypotheses under even more controlled conditions. We will highlight the need of such experimental strategies in the conclusion section:

Despite the difficulties associated with the statistically relevant sampling of atmospheric microorganisms, individual environmental factors have been started to be identified that control the concentration (Archer et al., 2019; Gusareva et al., 2019), diversity (Tong and Lighthart, 1997; Bryan et al., 2019) and selection (Smith et al., 2011; Joly et al., 2015) of atmospheric microorganisms. While it remains difficult to perform sampling with sufficiently high resolution and frequency, dedicated strategies should be developed to constrain the role of individual environmental parameters (e.g., UV light, temperature) for microbial diversity and survival in the atmosphere. Such studies should be accompanied by suitable lab or chamber studies under controlled conditions to test hypotheses that may be formed based on ambient observations.

4) The calculation on settling velocity of microbial cells does not take into account air currents. In addition, it is not impossible that some microbes or microbial spores have evolved for long-range dispersal by air – this has long been known for plant seeds, and there is mounting evidence of this for fungi (e.g. Borgmann-Winter et al. 2023, Ecology 104), therefore it is not unlikely that this is relevant for prokaryotes. Previous modelling work has suggested that long-range microbial dispersal is likely (Wilkinson et al. 2012 J. Biogeography 39).

**Author response**: We thank the referee for pointing out this aspect. We are aware of several studies that indicate long-range transport of bacteria. Such transport is included in the calculation of the atmospheric residence time as derived from the global model study by Burrows et al. (2009). This model takes into account both vertical and horizontal transport processes of air masses containing gases and particulate matter (including bacteria). The fact that the residence time is not a linear function of the settling velocity but is affected by numerous other factors, will be more clearly pointed out in Section 2.1 and in the caption of Figure S1:

Section 2.1 (new text in **bold**): In addition to gravitational settling, other particle properties and processes affect the atmospheric residence time, **including horizontal transport and** the ability to act as cloud condensation or ice nuclei. Burrows et al. (2009a) demonstrated that bacteria acting as cloud condensation nuclei (CCN) **are generally deposited faster and, thus,** have a global atmospheric residence time that is approximately half as long as bacteria not activated into droplets. **Therefore, the distances that CCN- and/or IN-active bacteria travel in the atmosphere are generally comparably short.**

Figure S1 (new text in **bold**): $\tau_{atmos}$ denotes the mean atmospheric residence times for CCN-active bacteria as  **derived from a global atmospheric model study** by Burrows et al. (2009).

**References**

[revised manuscript text omitted]

---

## Author Comment (AC2)

**Author response to comments by Referee #2**

All referee comments are shown in black, our author responses in blue; suggested new manuscript text is indicated in red with text suggested to be removed in *red italics*.

**General comments:**

Ervens et al. presents a thoughtful, well-written piece summarizing previous, and motivating future, research on microorganisms in the atmosphere. The detailed figures were especially informative and effectively conveyed the concepts discussed throughout the article. While some considerations are not wholly original, they are clearly and concisely encapsulated here.

**Author response:** We thank the referee for their positive and constructive comments. We agree that not all concepts presented in our article are completely new. However, it was not the primary motivation of this 'Ideas & Perspectives' article to present entirely new findings but instead to synthesize current knowledge from the intersections of atmospheric chemistry, microphysics and biology. With the recent growth of interest in atmospheric biology - both in the fields of atmospheric sciences and biogeosciences - we seek with this article to put (more or less) well-known facts into a broader context.

**Specific comments:**

Page 3, Lines 57-59: The sentence discussing settling velocity is a bit unclear. Consider replacing "particle size" with "particle diameter". Are you assuming that doubling the number of cells would double the particle diameter? What about in the case of high RH or a cloud droplet where a second cell may just displace water (cf. Figure 3)?

**Author response:** We agree with the referee that this sentence may have oversimplified the relationship between number of cells and particle size or even surface ($\propto v_t$). Two cells (even if of identical sizes) may not double the surface of the particle due to more compact geometric arrangement. It may lead, however, to more water uptake since more hygroscopic mass will lead to more water uptake. We modified the sentence as follows:

*The settling velocity of particles approximately scales with the square of particle size. thus, doubling the number of cells (of same size) in a single particle may decrease their settling time by a factor of 4 (Section S1.1, supplemental information).*

The presence of more than a single cell in a particle leads to a larger particle. However, the resulting total particle surface area might not scale proportionally with the number of cells since the particle shape and total volume might be mostly determined by the hydration shell.

Page 3, Lines 63-64. Please provide a reference for these statements. It may be appropriate to cite Fankhauser et al. (2019) who were among the first to suggest that microbes were physically isolated from one another in the atmosphere.

**Author response:** We realized that the underlying assumptions for the second sentence were not fully clear. We added appropriate references and an expanded text how to derive the fraction of bacteria in CCN populations. We would like to point out that these conclusions were not unique to Fankhauser et al. Instead, we cite Ervens and Amato (2020) where it is explicitly stated that "bacteria are unevenly distributed among cloud drop populations as statistically only 1 in ∼10 000 droplets may contain a single bacterial cell", together with some basic numbers on particle concentrations and sizes.

*Fewer than 1 out of 1000 atmospheric aerosol particles contain a bacteria cell. Accordingly, it may be concluded that the number fraction of bacteria-containing droplets is on a similar order of magnitude.*

Atmospheric concentrations of bacteria cells are typically in the range of 0.001 - 0.1 cells $cm_{air}^{-3}$ (Burrows et al., 2009; Després et al., 2012) with typical sizes on the order of 100 nm - 1 $\mu$m (Sattler et al., 2001; Pöschl and Shiraiwa, 2015). The total atmospheric number concentration of aerosol particles of such sizes ('fine particles') ranges from $10^3$ - $10^5$ particles $cm_{air}^{-3}$ (Seinfeld and Pandis, 2006). The comparison of these numbers reveals that bacteria comprise $\ll$1% of all atmospheric aerosol particles. A cloud droplet forms by water vapor condensation on an individual particle, i.e. on a cloud condensation nucleus (CCN) that is typically in the size range of fine particles. The fact that the bacteria number concentration is much smaller than the total CCN concentration in the atmosphere led Ervens and Amato (2020) to conclude that only 1 out of ∼10000 cloud droplets contains a bacteria cell.

Page 6, Section 2.3: This section assumes that microorganisms are metabolically active in the atmosphere. The article would benefit from a brief discussion of dormancy, in relation to this and other stressors.

**Author response:** We thank the referee for this suggestion. Indeed, we imply that bacteria are metabolically active in this section. To clarify this caveat, we add the following text at the beginning of

this section to point out the different levels of activity, despite very little data on this on atmospheric microorganisms:
In the atmosphere, bacteria cells may exhibit different levels of metabolic activity, which range from mere survival strategies, i.e., activity focused solely on repairing cellular damage, to dormancy, during which cells sustain their essential biological functions, to growth and multiplication as the most energy-intensive activities (Price and Sowers, 2004). Cells may become dormant under water-limited conditions (Haddrell and Thomas, 2017; Smets et al., 2016) or due to other stressors (Šantl-Temkiv et al., 2022). In cloud water, Sattler et al. (2001) observed cell activity at 0°C compatible with cell growth, whereas dormancy was observed outside clouds (Smets et al., 2016). Given that particles (including bacteria calls) only spend a fraction of their time inside clouds ((Ervens and Amato, 2020)), it can be, thus, expected that many bacteria may be dormant for long period of their atmospheric residence time. Dormancy has been shown in other environments to be an efficient response to harsh conditions and ultimately being beneficial for survival (Jones and Lennon, 2010).

In the conclusion section, we modified the following sentence:
Similarly, the rationale for exploring biodegradation rates in cloud water could be extended from focusing on potential impacts on chemical budgets to consequences of limited nutrient availability on **levels of metabolic activity, including dormancy,**  starvation and survival.

pH response (Author Response to Referee #1): The inclusion of a new subsection on effect of pH is appreciated. It is suggested to add additional commentary in light of work by Liu et al. (2023, ACP) whose laboratory experiments reported on the effects of pH (in combination with light exposure) on bacterial survival.
**Author response:** We thank the referee for reminding us of the study by Liu et al. (2023). In addition, to the text we suggested in our response to Referee 1, we will add:
Liu et al. (2023) found different trends when they examined the pH dependence of the survival and biodegradation rates of two strains of *Enterobacter* bacteria isolated from ambient air in a polluted environment: The showed that in the presence of light, the survival rate decreased in particular at pH $\leq 5$. These trends may point to different sensitivities of this particular bacteria type to pH, as compared to the responses by bacteria in cloud water (Vaïtilingom, 2013). The concurrent responses to low pH and the presence of sunlight may suggest some photolytic or photochemical mechanism that influences the biodegradation activity.

**Technical corrections:**
Page 2, Line 29: The word "role" is written twice.
Page 4, Line 79: Missing period after closed parenthesis and "Novel".
Page 4, Line 88: Extraneous closed parenthesis before comma.
Page 5, Line 108: Extraneous period between times and during.
**Author response:** Thank for pointing out these typos. They will be all corrected in the revised manuscript.

**References**

Burrows, S. M., Butler, T., Jöckel, P., Tost, H., Kerkweg, A., Pöschl, U., and Lawrence, M. G.: Bacteria in the global atmosphere – Part 2: Modeling of emissions and transport between different ecosystems, Atmospheric Chemistry and Physics, 9, 9281–9297, https://doi.org/10.5194/acp-9-9281-2009, 2009.

Després, V. R., Huffman, J. A., Burrows, S. M., Hoose, C., Safatov, A. S., Buryak, G., Fröhlich-Nowoisky, J., Elbert, W., Andreae, M. O., Pöschl, U., and Jaenicke, R.: Primary biological aerosol particles in the atmosphere: a review, Tellus B, 64, 15 598, https://doi.org/10.3402/tellusb.v64i0.15598, 2012.

Ervens, B. and Amato, P.: The global impact of bacterial processes on carbon mass, Atmospheric Chemistry and Physics, 20, 1777–1794, https://doi.org/10.5194/acp-20-1777-2020, 2020.

Haddrell, A. E. and Thomas, R. J.: Aerobiology: Experimental Considerations, Observations, and Future Tools, Appl. Environ. Microbiol., 83, https://doi.org/10.1128/AEM.00809-17, 2017.

Jones, S. E. and Lennon, J. T.: Dormancy contributes to the maintenance of microbial diversity, Proc. Natl. Acad. Sci. U.S.A., 107, https://doi.org/10.1073/pnas.0912765107, 2010.

Liu, Y., Lee, P. K. H., and Nah, T.: Emerging investigator series: Aqueous photooxidation of live bacteria with hydroxyl radicals under clouds-like conditions: Insights into the production and transformation of

biological and organic matter originating from bioaerosols, Environmental Science: Processes & Impacts, https://doi.org/10.1039/D3EM00090G, 2023.

Pöschl, U. and Shiraiwa, M.: Multiphase Chemistry at the Atmosphere–Biosphere Interface Influencing Climate and Public Health in the Anthropocene, Chemical Reviews, 115, 4440–4475, https://doi.org/10.1021/cr500487s, 2015.

Price, P. B. and Sowers, T.: Temperature dependence of metabolic rates for microbial growth, maintenance, and survival, Proc. Natl. Acad. Sci. U.S.A., 101, 4631–4636, https://doi.org/10.1073/pnas.0400522101, 2004.

Šantl-Temkiv, T., Amato, P., Casamayor, E. O., Lee, P. K. H., and Pointing, S. B.: Microbial ecology of the atmosphere, FEMS Microbiology Reviews, p. fuac009, https://doi.org/10.1093/femsre/fuac009, 2022.

Sattler, B., Puxbaum, H., and Psenner, R.: Bacterial growth in supercooled cloud droplets, Geophysical Research Letters, 28, 239–242, https://doi.org/10.1029/2000GL011684, 2001.

Seinfeld, J. H. and Pandis, S. N.: Atmospheric Chemistry and Physics - From air pollution to climate change, John Wiley & Sons, Inc., Hoboken, New Jersey, 2nd edn., 2006.

Smets, W., Moretti, S., Denys, S., and Lebeer, S.: Airborne bacteria in the atmosphere: Presence, purpose, and potential, Atmospheric Environment, 139, 214–221, https://doi.org/10.1016/j.atmosenv.2016.05.038, 2016.

Vaïtilingom, M.: Potential impact of microbial activity on the oxidant capacity and organic carbon budget in clouds, Proceedings of the National Academy of Sciences USA, 110, https://doi.org/10.1073/pnas.1205743110, 2013.

---

## Author Response (AR3)

Response to the editor comment:

**Editor comment:** Thank you for the comprehensive responses to Referee comments. I feel that the revised manuscript is publishable but I was surprised at the choice to not cite Fankhauser et al. (2019, https://pubs.acs.org/doi/full/10.1021/acsearthspacechem.9b00054) in the review as recommended. In my opinion it can be cited alongside Evrens and Amato (2020) as noted in the response letter; excluding it from the citations list of an Ideas and Perspectives manuscript strikes me as unnecessarily excluding the important work of talented young scientists who are making key contributions to this important field.

**Author response:** We thank the editor for accepting the revision of our manuscript. We are happy the last final change and added the citation to Fankhauser et al. (2019) at two places (bottom of p. 3/ top of p. 4) (addition in **bold**):

**Since single bacterial cells are often similar in size to these particles, it is likely that each particle hosts only one cell (Fankhauser et al., 2019).** The fact that the bacteria number concentration is much smaller than the total CCN concentration in the atmosphere led **Fankhauser et al. (2019) and** Ervens and Amato (2020) to conclude that only 1 out of ∼10000 cloud droplets contains a bacteria cell.

**References**

Ervens, B. and Amato, P.: The global impact of bacterial processes on carbon mass, Atmospheric Chemistry and Physics, 20, 1777–1794, https://doi.org/10.5194/acp-20-1777-2020, 2020.

Fankhauser, A. M., Antonio, D. D., Krell, A., J., A. S., Banta, S., and Mc Neill, V. F.: Constraining the Impact of Bacteria on the Aqueous Atmospheric Chemistry of Small Organic Compounds, ACS Earth Space Chem, 3, 1485–1491, https://doi.org/10.1021/acsearthspacechem.9b00054, 2019.